# An Unprecedented Bloom of Oceanic Dinoflagellates (*Karenia* spp.) Inside a Fjord within a Highly Dynamic Multifrontal Ecosystem in Chilean Patagonia

**DOI:** 10.3390/toxins16020077

**Published:** 2024-02-02

**Authors:** Ángela M. Baldrich, Patricio A. Díaz, Sergio A. Rosales, Camilo Rodríguez-Villegas, Gonzalo Álvarez, Iván Pérez-Santos, Manuel Díaz, Camila Schwerter, Michael Araya, Beatriz Reguera

**Affiliations:** 1Centro i~mar, Universidad de Los Lagos, Casilla 557, Puerto Montt 5480000, Chile; angela.baldrich@ulagos.cl (Á.M.B.); patricio.diaz@ulagos.cl (P.A.D.); camilo.rodriguez@ulagos.cl (C.R.-V.); ivan.perez@ulagos.cl (I.P.-S.); camila.schwerter@ulagos.cl (C.S.); 2Centre for Biotechnology and Bioengineering (CeBiB), Universidad de Los Lagos, Casilla 557, Puerto Montt 5480000, Chile; 3Programa de Doctorado en Biología y Ecología Aplicada, Universidad Católica del Norte, Coquimbo 1780000, Chile; sergiorosalesg@gmail.com; 4Facultad de Ciencias del Mar, Departamento de Acuicultura, Universidad Católica del Norte, Coquimbo 1780000, Chile; gmalvarez@ucn.cl; 5Centro de Investigación y Desarrollo Tecnológico en Algas (CIDTA), Facultad de Ciencias del Mar, Universidad Católica del Norte, Coquimbo 1780000, Chile; mmaraya@ucn.cl; 6Centro de Investigación Oceanográfica COPAS Sur-Austral y COPAS COASTAL, Universidad de Concepción, Concepción 4030000, Chile; 7Programa de Investigación Pesquera, Universidad Austral de Chile, Puerto Montt 5480000, Chile; manueldiaz@uach.cl; 8Centro Oceanográfico de Vigo, Centro Nacional Instituto Español de Oceanografía (IEO-CSIC), Subida a Radio Faro 50, 36390 Vigo, Spain

**Keywords:** *Karenia* species, Patagonian fjords, harmful algal blooms, fish killers, climate variability

## Abstract

At the end of summer 2020, a moderate (~10^5^ cells L^−1^) bloom of potential fish-killing *Karenia* spp. was detected in samples from a 24 h study focused on *Dinophysis* spp. in the outer reaches of the Pitipalena-Añihue Marine Protected Area. Previous *Karenia* events with devastating effects on caged salmon and the wild fauna of Chilean Patagonia had been restricted to offshore waters, eventually reaching the southern coasts of Chiloé Island through the channel connecting the Chiloé Inland Sea to the Pacific Ocean. This event occurred at the onset of the COVID-19 lockdown when monitoring activities were slackened. A few salmon mortalities were related to other fish-killing species (e.g., *Margalefidinium polykrikoides*). As in the major *Karenia* event in 1999, the austral summer of 2020 was characterised by negative anomalies in rainfall and river outflow and a severe drought in March. *Karenia* spp. appeared to have been advected in a warm (14–15 °C) surface layer of estuarine saline water (S > 21). A lack of daily vertical migration patterns and cells dispersed through the whole water column suggested a declining population. Satellite images confirmed the decline, but gave evidence of dynamic multifrontal patterns of temperature and chl *a* distribution. A conceptual circulation model is proposed to explain the hypothetical retention of the *Karenia* bloom by a coastally generated eddy coupled with the semidiurnal tides at the mouth of Pitipalena Fjord. Thermal fronts generated by (topographically induced) upwelling around the Tic Toc Seamount are proposed as hot spots for the accumulation of swimming dinoflagellates in summer in the southern Chiloé Inland Sea. The results here provide helpful information on the environmental conditions and water column structure favouring *Karenia* occurrence. Thermohaline properties in the surface layer in summer can be used to develop a risk index (positive if the EFW layer is thin or absent).

## 1. Introduction

Climate change has become a major issue for increasingly exploited aquatic ecosystems, particularly fjords, coastal lagoons, and other semi-enclosed systems [1,2]. These systems are highly vulnerable because they are subject to complex ocean–atmosphere–biological interactions of multiple scales that control the response of planktonic communities including harmful algal blooms (HABs) [3,4]. High biomass (>10^6^ cells L^−1^) blooms of fish-killing microalgal species in fjordic systems in Northern Europe and Southern Chile have drawn considerable attention in the last two decades due to the severity of their socio-economic impacts on the caged salmon industry [5]. The causative microalgal genera belong to a variety of taxonomic groups including raphydophytes (e.g., *Chattonella, Heterosigma*), dinoflagellates (e.g., *Karenia*, *Margalefidinium*), dictyochophyceans (e.g., *Pseudochattonella*, *Vicicitus*), and haptophyceans (e.g., *Chrysochromulina*, *Prymnesium*) [6]. Notwithstanding their socio-economic impacts, knowledge of the chemistry of the bioactive compounds associated with these blooms and their multiple effects and mechanisms of action is much further behind than that of shellfish toxins [7,8].

Dinoflagellate species of the genus *Karenia* G. Hansen and Moestrup (formerly *Gymnodinium*), associated with mass mortalities of cultivated oysters and fish in Japan in the 1930s (*K. mikimotoi*) [9] and marine fauna in the Gulf of Mexico in the 1940s (*K. brevis*) [10], were the first recognised fish-killing agents reported in the HAB literature. *Karenia* species, nine of which have been included in the reference list of noxious species [11], are widespread from subtropical (Northern Gulf of Mexico) to Austral and Boreal seas [12,13].

Blooms with a mixture of several Kareniaceae are not uncommon [14,15,16,17] and noxious effects reported are species’ strain-dependent and may include neurotoxic shellfish poisoning (NSP or brevetoxicity), breathing difficulty by the inhalation of sea spray-borne toxins, anoxic events, and mass mortalities of marine fauna and aquaculture resources [13,18]. To date, the small list of unambiguously confirmed bioactive compounds of *Karenia* species include brevetoxins in *K. brevis* from Florida [19], gymnodimines from *K. selliformis* in New Zealand [20,21], brevisulcenals in *K. brevisulcata* [22], gymnocines in *K. mikimotoi* [23], polyunsaturated fatty acids (PUFAs) [24], sterols [25], and other toxins of unknown mechanisms of action [26]. Gymnodimine-A is the only certified toxin standard commercially available, and there is no assay developed that covers the whole array of bioactive compounds [27,28].

*Karenia* species are naked dinoflagellates with high morphological variability, including small cell formation; cells become deformed and difficult to identify with light microscopy in phytoplankton samples with standard Lugol’s fixative [12,14]. In recent years, the application of molecular probes has revealed previous misidentifications of species morphologically similar to *K. brevis*, such as some Pacific phylotypes of *K. papilionacea*, during blooms of *Karenia* in New Zealand and Japan [15,29]. From all the above, it is easy to understand the uncertainties faced by experts when trying to identify the precise species and bioactive compounds and the mechanisms of action responsible for particular mass mortalities of marine organisms.

*Karenia* species are extremely versatile when it comes to nutritional sources and light preferences. These mixoplanktonic (photo-osmo-phagotrophs sensu [30]) dinoflagellates are able to perform photosynthesis (photo-) with their constitutive haptophyte-like plastids, take up dissolved organic matter (osmo-), and eat small cyanobacteria (phagotroph). They thrive on regenerated N sources, such as ammonia, urea, and polyamines from decomposing diatom blooms or animal waste [31] and have been found to feed on cyanobacteria (*Synechochoccus* spp.) in laboratory cultures [32]. Phototactism and photoadaptation allow them to grow in low-light environments (within the pycnocline) or endure strong radiation near the surface as they are protected by their accessory pigments. Unlike other dinoflagellates, Kareniaceae lack peridinin and have a distinct combination of fucoxanthin-related accessory pigments [33]. This characteristic has facilitated the development of algorithms for the teledetection of *Karenia* populations in advanced operational oceanography programmes [34], provided they reach a high cell density (>10^6^ cell L^−1^) and are located near the sea surface in light hours (reviewed in [13,33]).

Bloom dynamics of *Karenia brevis* in the Gulf of Mexico and *K. mikimotoi* in European Atlantic and Korean shelf waters have been well studied [31,35]. Steidinger et al. [36] pointed to the analogies between the different growth phases (lag, exponential, and stationary) observed in monoalgal cultures and those in mid-shelf occurring dinoflagellate populations such as *K. brevis* (i.e., initiation, exponential growth, and stationary phase), with an additional transport phase at the end of their growth season [35,37]. Bloom initiation takes place when sparse overwintering populations aggregate in well-established late spring–summer pycnoclines [31]. High biomass blooms (>10^6^ cells L^−1^) appear aggregated in tidal/upwelling/estuarine frontal areas (e.g., the tidal Ushant front in Brittany) [38], and wind-driven circulation has been identified as the key factor controlling (i) large distance longshore transports, entrained in gyres and eddies such as the coastal jet between England and Southern Ireland [39] and the Gulf Stream around shelf waters off eastern Florida and Southeast USA [35], and (ii) cross-shelf transport, which may result in the advection of dense blooms to coastal tourist areas and aquaculture sites.

At the microscale, water column stratification patterns play an important role in controlling *Karenia* swimming behaviour, daily vertical migration, and photoadaptations [40,41]. Exceptional blooms (>10^7^ cells L^−1^) of *Karenia* have been associated with very hot and dry summers and wind anomalies favouring the approach of shelf populations to the coast. Oceanic populations of *K. brevis* have also been described in oligotrophic waters of the Gulf of Mexico. Therefore, despite their moderate growth rate (µ_max_~0.4 d^−1^), *Karenia* species have functional traits allowing them to adopt various strategies to persist in the system and develop thick blooms when the appropriate environmental window occurs. Fish kills are possible with concentrations exceeding 10^5^ cells L^−1^ [13].

### Background of Karenia Harmful Events in Patagonia

The fjord and channel system in Western Patagonia (Southern Chile) (~240,000 km^2^) is one of the largest in the world [42]. The geomorphology and hydrodynamics of these systems are extremely complex, exhibiting an array of microenvironments with site-specific characteristics directly affecting the composition and dynamics of phytoplankton assemblages [43,44]. Strong and permanent vertical salinity gradients are shaped by seasonal and latitudinal patterns of freshwater inputs from rivers, ice melting, and rainfall [43,45,46]. The two-layered estuarine-like circulation is driven by salinity and characterised by outflowing surface water and inflowing bottom water (with restrictions from a sill) separated by an intermediate layer [47]. Environmental variability is maximal at the surface (~top 10 m) estuarine water (EW) layer. This surface layer has been subdivided, on the basis of salinity into three ranges labelled as fresh water (FW, salinity < 11), estuarine fresh water (EFW, salinity: 11–21), and estuarine saline water (ESW, salinity: 21–31) [48]. An intermediate layer of modified subantarctic water (MSAAW, salinity: 31–33) separates the ESW from a saltier, more uniform subantarctic water layer (SAAW, salinity: >33), which reaches a depth of 150 m [49,50].

Pitipalena (~43° S) (Aysén region), a 22 km long semi-enclosed system with one connection to the Corcovado Gulf, is a northwestern Patagonian fjord located within the Pitipalena-Añihue Marine Protected Area (D.S No. 13 Ministry of the Environment—MMA) [51] (Figure 1). High freshwater inputs come from riverine inflows and rainfall (~4500 mm yr^−1^). The main river flowing into the fjord, the Palena River (average river discharge 800 m^3^s^−1^), is located at the very mouth of the fjord [52,53]. This distinct feature affects the site-specific hydrodynamics of Pitipalena Fjord, including water residence time (~200 days) [54,55], and the advection and retention of phytoplankton populations, including HABs [56].

Time series analyses of meteorological and hydrographic conditions from Northern Patagonia (41° S to 46° S) have shown a decreasing trend in rainfall and river outflow rates in recent decades [57,58]. These climate anomalies affect the exchange of phytoplankton populations, including harmful algal blooms, between upwelled waters from the open Pacific (southernmost limit of the Humboldt Current upwelling system) and Corcovado Gulf through the Boca del Guafo channel [59,60,61]. Additional local upwelling is generated by wind interactions with the local topography, in particular around the Tic Toc Seamount, leading to a complex array of eddies and oceanic fronts [60]. These conditions make Corcovado Gulf a highly productive area, recognised as one of the most important feeding grounds for the blue whale in the Southeast Pacific Ocean [62,63,64].

Blooms of *Karenia* species in Chilean Patagonia have been mostly observed in offshore areas [28]. The first major fish-killing bloom of a *Karenia* species (identified as *Gymnodinium* sp.) in 1999 occurred when dense populations (>8 × 10^6^ cells L^−1^), presumably advected from oceanic waters through Boca del Guafo, reached a salmon farm area in the southeast part of Chiloé Island that faces the Corcovado Gulf (Los Lagos region) [65]. Shortly after, a brown tide of the same “*Gymnodinium*” killed invertebrates and fish in the exposed Magellanic fjords [66].

In recent years, fish-killing blooms of *Karenia* species have been observed in open shelf waters off Los Lagos and Magallanes, two of the three southernmost regions of Chilean Patagonia. In January 2017, around 170,000 salmons died while in well-boat transit between Magallanes (~54° S) and Los Lagos (~41° S). Water samples taken a few days later in 10 sampling stations between the Gulf of Penas (~47° S) and Moraleda Channel (~44° S) showed that moderate densities of *Karenia* spp. (>65 × 10^3^ cells L^−1^), including *K. brevis*, *K. papilionacea*, *K. mikimotoi*, *K. brevisulcata*, and *K. bidigitata*, were detected only in high-salinity (>34) oceanic waters [17]. The same pattern was observed between February 2017 and March 2018 [67]. The latest large-scale *Karenia* (*K. selliformis*) fish-killing event in Northwestern Patagonia (37° S to 45° S) was in summer 2018 [27]. It has been suggested that populations of *Karenia* spp. in Southern Chile build up in the open Pacific coastal waters and are presumably advected (physical transport) into the adjacent inland waters [17,66]. By the end of summer (18–19 March) 2020, an unprecedented bloom of *Karenia* was accidentally detected in samples from a 24 h study (focused on *Dinophysis* spp.) at a fixed station in Pitipalena Fjord (43°47′ S–72°56′ W) (Figure 1).

Despite increased reporting of *Karenia* spp. events and their severe impacts on wildlife and salmon aquaculture in Chilean Patagonia, little is known about the environmental factors contributing to their occurrence in fresher inland waters. Nevertheless, exchanges of oceanic front populations (including fish-killing species) with the outer reaches of the fjords are expected to be enhanced in parallel with the trend of increased sea surface salinity (SSS) and lower riverine outflow in Northwest Patagonian summers [57,59].

The objectives of this work were to (i) describe environmental conditions and water column structures related to the vertical distribution of *Karenia* spp.; (ii) explore the potential causes of a moderate (max~10^5^ cells L^−1^) 2020 *Karenia* event in an unusual location, and (iii) review limitations when trying to monitor and give an early warning of *Karenia* spp.

## 2. Results

### 2.1. Meteorological and Hydrographical Conditions

Summer 2020, in particular March, was extremely dry. Monthly rain values of 10.9 mm, 10.8 mm, and 7.7 mm in January, February, and March, respectively, were 5–10% of the average from the previous 17 years (2003 to 2019) (Table 1). Values during the sampling days (March 18–19) (Figure 2A) were 0 and 0.2 mm, respectively.

Daily average outflow of the Palena River in 2020 showed characteristic low values in summer, similar to the historic mean in January and February, but relevant negative anomalies were observed during most of March (Table 1). The lowest rate (anomaly of −446.3 m^3^seg^−1^) was observed on March 9, and it was close to the minimum (328.46 m^3^seg^−1^) that was observed during the 24 h sampling (March 18 to 19) (Figure 2B).

Hydrological measurements at the fixed station showed a sharp thermal stratification within a warm (>14 °C), thin surface layer. This layer, of variable depth (0.5–2 m) associated with the semidiurnal tidal cycle (Figure 3A), was 2 m deep during the first eight hours of the study when the EFW (S: 11–21) was practically absent. After 03:00 h, a deepening of the EFW layer followed, with a maximum of 2 m at 10:00 h. Below the EFW, the layer of ESW (S: 21–31) extended down to 10–20 m above the modified subantarctic water (MSAAW) (Figure 3B,C). Higher values of buoyancy (Brunt Väisälä) frequency (~70 cycles/h) occurred in the first 2 m during ebb tide, with a maximum of 125 cycles/h at the very surface (Figure 3D).

The distribution of inorganic nutrients paralleled the semidiurnal tide. High levels of nitrates, harmonically distributed, showed a mean of 8.00 ± 4.17 µmol L^−1^ and a maximum of 13.84 µmol L^−1^ at 20 m (Figure 4A). Low values of nitrites (0.41 ± 0.13 µmol L^−1^) were homogeneously distributed (Figure 4B). Phosphates (mean 1.13 ± 0.16 µmol L^−1^), ranging from undetectable levels at the surface to a maximum (1.80 µmol L^−1^) at the phosphocline at 15 m, showed a harmonic distribution similar to that of nitrates (Figure 4C). Silicates (8.69 ± 2.8 µmol L^−1^) were highest at ebb tide in the top 2 m (~10 µmol L^−1^) (Figure 4D).

### 2.2. Karenia Cell Morphology

Lugol´s fixed cells of *Karenia* were 20.7 ± 2.1 µm long (L) (range: 17–26 μm; n = 50) and 17.2 ± 1.7 µm wide (W) (range: 14–21 μm; n = 50) (Figure A1). Cells were dorso-ventrally compressed. The epicone was hemispherical in some specimens and slightly pointed in others. Likewise, the bilobulated hypocone was almost hemispherical in some specimens and slightly concave in others. Nuclei were observed extending through the two lobular halves in the hypocone (Figure A2).

### 2.3. Vertical Distribution of Karenia sp. and Toxins

Changes in the vertical distribution of *Karenia* populations during the 24 h sampling followed the excursions of the isopycnals related to the semidiurnal tidal signal (Figure 5A,B). Maximal cell numbers (1.4 × 10^5^ cells L^−1^) were observed at 20:00 h within the warm (>14 °C) surface (0–2 m) layer of ESW (S > 21). A sharp decline in *Karenia* cell numbers (6 × 10^3^ cells L^−1^) occurred at 6:00 h on March 19, coinciding with the ebbing tide and a 4 m deep surface layer of EFW. A second peak (5–7 × 10^4^ cells L^−1^) was observed from noon to 16:00 h on March 19. This second peak, again, appeared to coincide with the flooding tide, although the EFW occupied the top 2 m and the ESW extended from 2 to 8 m. All through the study, *Karenia* cell maxima were observed in the top 6 m within the ESW (21–31) and there was not a clear pattern of daily vertical migration. Targeted toxin analyses by LC-HRMS of plankton net tows collected every hour showed no traces of GYM-A in any of the 24 samples tested (Figure A3).

### 2.4. Satellite Observations

Satellite images were obtained from Copernicus Sentinel-2 and Sentinel-3 on March 26 (one week after the 24 h study), March 30, and April 8 (Figure 6). Satellite GHRSST images (Figure 6A–C) of sea surface temperature (SST) on March 26 showed a thermal front on the western side of the Tic Toc Seamount. On March 30, the most marked frontal area was observed in the interface between Moraleda Channel and the southern area of Corcovado Gulf. By April 8, the two fronts had weakened.

Natural colour images and chl *a* on the same dates (Figure 6D–I) showed that on March 26, the highest colour intensities were observed in Moraleda Channel and off the mouths of Pitipalena and Reloncaví fjords, limited by the eastern side of the Tic Toc Seamount. Five days later, on March 31, colour maxima were observed in the central region of Corcovado Gulf and Boca del Guafo. Finally, on April 8, the satellite image showed the highest (colour) concentration associated with a frontal feature extending from Corcovado Gulf to Boca del Guafo. Smaller scale patches were observed. Ocean colour images from Copernicus Sentinel-3 were filtered and processed to identify pigment signatures from specific taxa (Figure 6J–L). The results indicated a very low contribution of *Karenia* spp. (with distinct fucoxantin accessory pigments) to the total chlorophyll recorded in Corcovado Gulf and Moraleda Channel on the three occasions.

Weekly monitoring in salmon farm sites carried out by the Chilean National Fisheries and Aquaculture Service (SERNAPESCA) showed the occurrence of low densities of *Karenia* spp. (<10^4^ cells L^−1^) and high numbers (>10^5^ cells L^−1^) of *Lepidodinium chlorophorum* (Figure A4). Between 6 and 19 April, there were reports on the occurrence of *Margalefidinium* and *Lepidodinium* species. Two minor salmon mortality events were associated with a bloom of *Margalefidinium* (=*Cochlodinium*) *polykrykoides*. This dinoflagellate produces abundant mucilage that gets attached to the fish gills and suffocates them. This was co-occuring with diatoms (*Chaetoceroa cryophilus*, *Eucampia zodiacus*) and silicoflagellates (*Dyctyocha speculum*) species previously related to fish mortality in Patagonian salmon farms. The bloom of *C. polykrikoides* was pointed out as the cause of nearly one million salmon mortalities on a farm in Quellón (Southern Chiloé) on 8 April.

Green discolourations attributed to *Lepidodinium chlorophorum* extended from Southern Chiloé Island to the northern coasts of Aysén province. These *Lepidodinium* patches probably corresponded to the intense colouration on the satellite image from April 8. Neither monitoring centres nor salmon farmers or additional satellite information made special remarks about the occurrence of high densities of *Karenia*.

## 3. Discussion

Chilean Patagonia has a long history of shellfish poisoning events, but reports of fish kills are quite recent and linked to the onset of the salmon industry in 1982 [68]. Farmed salmon played the role of the caged canary in the mine, indicating the presence of species, often classified as *Gymnodinium* sp., that might have always been there. Alternative explanations could be (i) underreporting of unidentified *Gymnodinium* spp. until the first mass mortality occurred; (ii) growth enhancement of mixoplanktonic species, including *Karenia* spp., with increased input or organic nitrogenous compounds excreted by farmed fish; (iii) climate-related changes in water circulation approaching frontal populations to the fjords; or (iv) new introduction of *Karenia* spp. in the region by ballast waters and other ship vectors. From 1982 to the end of the century, fish kill events were caused by high biomass HABs of non-toxic species causing mechanical gill damage or abrupt changes in physico-chemical conditions (hyperoxygenation, anoxia, mucus secretion) common to any high biomass bloom. The only exception was a bloom of *Heterosigma akashiwo* in 1988 that killed several tonnes of salmon in Reloncaví Sound [69].

The first report of a large-scale *Karenia* fish-killing event in Southern Chile was in March 1999 [65] and the most affected salmon farm area (Quellón) was off the southwestern coast of Chiloé Island, limited to the south by Boca del Guafo and to the east by Corcovado Gulf. A second event, in January–February 2017, was mainly in offshore waters and affected wild fauna and well-boat salmon while in transport from Aysén to Los Lagos [17]. In the austral summer of 2018, a large-scale (hundreds of kilometres) bloom of Kareniaceae killed millions of pelagic and benthic marine animals on the western Patagonian coast [70]. The predominance of *K. selliformis* was confirmed [27]. By the end of summer 2020, a moderate-density population of *Karenia* spp. (>10^5^ cells L^−1^) was fortuitously detected in samples from Pitipalena Fjord collected during a 24 h study focused on *Dinophysis* [56]. Here, for the first time in Chile, the microscale distribution of *Karenia* spp. is related to water column structure and associated environmental conditions.

### 3.1. The Morphology and Toxic Potential of the Karenia Population in Pitipalena Fjord

*Karenia* spp. cells, including *K.* cf *selliformis*, observed in samples from this survey were deformed after preservation with acidic Lugol’s solution and detailed morphological descriptions were unfeasible. The low-resolution micrographs suggested the occurrence of at least two morphospecies. The observation of the cell nucleus extended through the left and right bottom lobes of the hypocone and the lack of gymnodimines in the cell extracts were characteristics of *Karenia selliformis*, as described in cultured strains from a previous mass mortality event in the region [28]. Measurements of that strain were a bit larger than those of *Karenia* spp. observed in this study. Even larger were the measurements reported for the specimens from the 1999 event. Nevertheless, cell size is not a very robust criterion; *Karenia* species are known to produce gametes about half the size of the vegetative cells [36]. Unfortunately, there are no molecular probes yet in Chile designed to tag different species of *Karenia;* identification with electron microscopy tools requires previous sample preservation with a glutaraldehyde solution.

The production of a variety of neurotoxic and hemolytic compounds by *Karenia* species is very variable at species and strain levels. A large amount of information on *K. brevis* blooms in Florida has shown a relation between cell concentrations and the severity of toxic impacts ranging from the detection of neurotoxic shellfish poisoning (NSP) (>10^3^ cells L^−1^) to respiratory irritations from aerosolised toxins and damage to wild fauna (>10^5^–10^7^ cells L^−1^) [13]. During the 1999 and 2017 mass mortalities in Chile, cell concentrations of *Karenia* exceeded 10^6^ cells L^−1^ and discoloured the water. In summer 2020, cell maxima of *Karenia* in Pitipalena (~10^5^ cells L^−1^) at the onset of the COVID lockdown were at the lower limit of cell densities of *K. brevis* affecting wild fauna. They probably corresponded to the last phase of a declining bloom, according to satellite data and monitoring cell counts from surrounding areas. Nevertheless, poor sampling resolution was clearly insufficient to follow pre- and post-bloom changes in cell densities. Isolated reports related damage in the fauna to fish killers other than *Karenia* spp.

The toxins gymnodymin A and C (GYM-A, GYM-C) have been well characterised in *K. selliformis* strains from New Zealand [20,21]. GYM-A was detected in plankton net-hauls collected in oceanic waters off Aysén province by Trefault et al. [71], suggesting that different species of *Karenia* are part of the autochthonous phytoplankton assemblages in Chilean Patagonia. This toxin is the only *Karenia*-related gymnodimine with commercially available certified standards. Nevertheless, no traces of GYM-A were found in the LC-MS analyses of 24 vertical net tow samples collected during the present survey.

Cultures of *K. selliformis* and other species associated with fish kills in Southern Chile were tested with a highly sensitive cytotoxic assay, the fish RTgill-W1 cell line-based assay [28]. This *K. selliformis* strain revealed a much higher icthyotoxic potency (8% gill cell viability) than other well-known fish killers, such as *Heterosigma akashiwo* (81%), and lysed cells had a stronger effect than whole intact cells. It is important to mention here that this cytotoxic assay estimated a similar toxic potential for *Heterosigma akashiwo* and for the non-toxic *Prorocentrum micans.* A hypothesis to explain these results was that the noxious effects of some fish killers were due to some kinds of inducible allelopathic compounds that are not produced in the absence of competitors and grazers. Chemical analyses of the same strains confirmed the absence of GYM-A as well as GYM-B and GYM-C but pointed to the presence of a couple of compounds with chromatographic behaviour close to brevenal and a high content of polyunsaturated fatty acids (PUFA) [27]. Brevenal is a brevetoxin antagonist identified in cultures of *K. brevis* from the Gulf of Mexico [22]. The brevenal/brevetoxin ratio has been found to affect the differences in toxic potential between strains. Some brevetoxin-related toxins can be expected to be found in the Chilean strains of *K. selliformis* containing brevenal-like compounds. This would help to explain the severe mortalities of wild fauna and caged salmon during the 1999 and 2017 *Karenia* cf *selliformis* outbreaks [17,65], despite the absence of GYMs [71]. The presence of allelopathic compounds that inhibited the growth of co-occurring diatoms and some dinoflagellates was confirmed by Clément et al. [65] during the 1999 event; the same compounds were harmless to co-occurring *Alexandrium catenella.* In the present study, *Karenia* spp. co-occurred with *Dinophysis* (*D. acuta* in the first few hours and *D*. *acuminata* most of the time) and ciliates (*Mesodinium* spp.) [56].

### 3.2. The Distribution of Karenia Cells and Behaviour

*Karenia* cells were distributed, throughout the 24 h observations, in the top 10 m, with the cell maxima above 4 m. In the first few hours of the survey, these maxima were associated with a warm (>14 °C) layer of ESW (S > 21) and flooding tides; the minima were associated with the irruption of the fresher EFW (S: 11–21) and flooding tides again 12 h later. We do not know if the population was in a growing phase or in the last days of its growth season and transported by surface currents.

There were no signs of daily vertical migration (DVM) in the *Karenia* population during the 24 h observations. Previous studies with *K. brevis* [41] and *K. mikimotoi* [72] have shown that during the day, *Karenia* cells aggregate at the surface, forming dense brown patches, and disperse downward throughout the water column at night. It has been suggested that the interaction between this swimming behaviour and local hydrographic characteristics facilitates the aggregation of *K. brevis* cells at certain depths for specific processes [41]. Similar behaviour has been observed in cells of *K. mikimotoi*. Cells aggregate at the surface during the day and migrate down to a 20–25 m depth at night to bottom layers where the concentration of resources is higher [72]. Vertical displacements of *K. mikimotoi* have been found to be determined by water column stability: migration occurs when the water column is well mixed or the stratification is weak. When stratification is stronger, the population maximum remains in the pycnocline [31]. During the 1999 *Karenia* event [65], later confirmed to be *K. selliformis*, cells were described to aggregate in dark patches in the morning and get dispersed from the surface to 10 m deep at night. Our observations confirmed a lack of DVM in *K.* cf *selliformis* in a highly stratified water column with a very shallow pycnocline, but cells were not aggregated but dispersed in the water column. A probable difference between the two scenarios was concerning resource availability. Inorganic and organics nutrients, probably depleted in shallow oceanic waters (the 1999 case), were plentiful in our study due to the Palena River outflow, and the population seemed to be in a more advanced phase, close to the decline. Unlike *Dinophysis* and ciliates, *Karenia* spp. was present between 0 and 12 m of depth, above and below the strong density gradients of the pycnocline.

Salinity has been considered a key abiotic factor in the control of *Karenia* bloom development, but values associated with field populations of *Karenia* and optimal conditions in laboratory experiments have yielded quite diverse results [73,74]. Indeed, blooms of *K. brevis* have been recorded in eastern and western Florida shelf waters and throughout the Gulf of Mexico associated with relatively high salinity values (maximum growth rate between 30 and 34) [18,75]. Nevertheless, field studies in the northern Gulf of Mexico since 1996 have reported frequent blooms of *Karenia* spp. associated with fresher waters (S < 25) [76]. Laboratory experiments conducted by the same authors with different strains of *K. brevis,* one from John Pass, Mexico Beach, and the other from Charlotte Harbor, Florida, showed strain-specific low salinity (20 to 25) and high salinity (37.5 to 45) ranges to be optimal salinity windows for growth. These authors also reported that the lowest bounds of salinity used to keep *K. brevis* in culture, between 17.5 and 20, raised questions about *Karenia* sp. salinity tolerance. In Western Europe, *K. mikimotoi* is considered a eurythermal and euryhaline species, and blooms of this species have been associated with estuarine fronts formed between river plumes and saltier shelf waters (reviewed in [77]).

In our study, the temperature of the warm surface layer was about the same as that reported by Clément et al. [65] during the first massive fish kill in the same area. That event was also associated with a drought (over previous 14 months) and SST-positive anomalies. A bifactorial (temperature and salinity) study with a local strain of *K. selliformis* showed that μ_max_ (0.41 d^−1^) was achieved with the combination 30/18 °C, and the minimum (0.04 d^−1^) with the combination 20/9 °C [27]. A distribution model of *Karenia* in relation to the Patagonian water masses was proposed with projections of a future climate warming scenario. In that model, optimum growth was observed within the modified subantarctic waters (MSAAW, S > 31) and the upper limit of the ESW range (21–31), i.e., in the interface between the ESW and the MSSAW water mass. Nevertheless, our study showed that inside the fjords, the upper limit of the MSAAW showed vertical displacements between 10 and 20 m of depth and a water temperature was ≤12 °C.

Several investigations have shown that salinities below 24 prevent the occurrence of *Karenia* spp. (reviewed in [76]), giving rise to the “24 salinity barrier” hypothesis [78]. Here, the natural barrier seems to be the isohaline of 21, i.e., the presence of EFW. In an earlier study, we concluded that the presence of EFW was a salinity barrier preventing the development of *D. acuta* in the northern Patagonian fjords [56]. During the present survey, *D. acuta* only appeared, briefly, when the warmer and saltier water (ESW) was at the surface. Interestingly, in shelf waters off Brittany and Southern Ireland, *D. acuta* and *K. mikimotoi* develop in stratified waters in summer and are transported in coastal jets associated with the Ushant (La Manche, France) and Celtic Sea tidal fronts [79].

### 3.3. Climate Variability and Local Hydrodynamics in Northern Patagonia and Karenia spp. Blooms

In recent decades, large-scale climate variability in Northwestern Patagonia has shown a declining trend in rainfall and river flushing rates [45,57,58]. This trend has been highlighted due to its consequences that result from the balance between several processes of different scales forcing the system [3,4,80]: (i) the glacial fjords paradox means that hotter springs cause a rise in cold freshwater inputs from ice melt; (ii) a draught-driven rise in surface salinity creates favourable conditions in summer for the development of shelf stenohaline phytoplankton species that are restrained by low salinity—this is reflected in the thinning or disappearance of the FW layer, S < 11, in the outer reaches of the fjords; and (iii) the estuarine fronts move shore-wards and the entrainment of oceanic waters and phytoplankton species is promoted. The high heterogeneity in SST and colour distribution in the satellite images confirmed the dynamic exchange between open Pacific waters and Corcovado Gulf through Boca del Guafo and Moraleda channels described in previous studies (Figure 1) [48,81]. Here, frontal regions of various kinds were observed, presumably generated by the upwelling of saltier, nutrient-rich waters from the Pacific Ocean. These waters are at the southern limit of the Humboldt Current upwelling system, which, in the last few decades, has shown a poleward shift from 40 to ~42–43° S [82,83]. Additional fronts are formed by topographically induced upwelling around the Tic Toc Seamount [83,84] and the estuarine fronts in the interphase between the fjords outflow and the saltier waters of the Inland Sea. Pitipalena Fjord is at the same latitude (43.7° S) as Boca del Guafo. This position facilitates the advection to the fjord of oceanic phytoplankton species aggregated in frontal areas. Recently, in-situ measurements from an acoustic Doppler current profiler (ADCP) moored on the Guafo channel provided evidence of an eastward flow of oceanic water masses [85]. These oceanic waters (SAAW and ESSW) were identified in the past as the potential vectors of *Karenia* spp. into the Chiloé Inland Sea [65].

Wind-driven advection of adjacent oceanic waters (Corcovado Gulf and Moraleda Channel, Figure 1) to Pitipalena Fjord has been highlighted as a key factor in understanding HABs dynamics in the region [86]. Furthermore, reports from the *Alexandrium catenella* surveillance, detection, and control programme from SERNAPESCA and the Chilean Fisheries Development Institute (Instituto de Fomento Pesquero—IFOP) provided evidence of the presence of *Karenia* spp. and *Lepidodinium chlorophorum* in salmon farms from the Los Lagos and Aysén regions from March to the first days of May 2020 (data available at http://www.sernapesca.cl/programas/programa-alexandrium-catenella; accessed on 5 March 2023). Cell densities reported for *Karenia* were below 10^4^ cells L^−1^ and some salmon mortalities were related to *Margalefidinium polykrikoides*. Further satellite observations by other researchers concluded that the intense colour in early April corresponded to *L. chlorophorum* [87].

The co-occurrence of *L. chlorophorum* and *K. mikimotoi* blooms is commonplace in the Bay of Biscay (France) [77]. Studies from that region have shown that *L. chlorophorum* is a green dinoflagellate able to produce high biomass blooms, discolouring the sea with an intense emerald green appearance, detectable by satellite images [88], and strong chl *a* anomalies [77,89]. Long-term observations (1998–2012) showed that the two species, which share similar life cycles and behavioural (mixotrophic feeding and DVM) traits, occupy similar niches and grow as excluding competitors. Blooms of the two species may occupy vast extensions in the plumes of the Loire and Vilaine rivers and the stratified shelf area of the Western English Channel. The two species have been related to advection from frontal areas to the coastal aquaculture sites.

In the present study, most likely, *L. chlorophorum* outcompeted *Karenia* spp. in some areas of the Corcovado Gulf. An intriguing question is why a moderate *Karenia* spp. bloom (>10^5^ cells L^−1^) was only detected in the outer reaches of Pitipalena on 18–19 March 2020. Our results showed the occurrence of *Karenia* cells aggregated in a narrow and highly stratified warm surface layer of ESW that varied in response to the semidiurnal tide. A particular feature of Pitipalena Fjord is that its main freshwater input, the Palena River, flows into the fjord’s mouth. This freshwater flow could act as a barrier, preventing the entrance of stenohaline species from oceanic waters advected through the Boca del Guafo channel. We carried out an early intensive sampling in Pitipalena, including two acoustic Doppler profilers (ADCP) moored in the fjords’ mouth [52]. Current measurements from that cruise showed a complex surface circulation, which is illustrated here in a simplified diagram (Figure 7). Due to its peculiar coastline, the tidal flow entering the south coast of the central area (Ensenada Las Islas) generates an anticyclonic eddy in the interior of the fjord. The river outflow, pushed into the fjord by the tide, is entrained in the eddy, and hours later reaches the northern part of the channel. This modeled circulation is the likely cause of the time lag between flood tide and the detection of the silicate’s maximum. In contrast to what is expected, silicate maxima appeared at ebb tide, i.e., a 6 h time lag, which is the time it takes a particle entrained in the eddy to reach the sampling station.

### 3.4. Management Considerations

Precipitation deficits, reduction in river discharge into coastal waters, increases in solar radiation, and, therefore, an increase in sea surface temperature and upwelling-favourable winds offshore are the future projections for Northern Patagonia [58,90]. These local characteristics coupled to large scale atmospheric circulation modes (i.e., El Niño Southern Oscillation—ENSO—and the positive phase of the Southern Annular Mode—AM) may result in a higher frequency of extreme droughts and environmental conditions, triggering exceptional fish-killing HAB events such as those that occurred in 2016 [28,91].

In the last two decades, drier summers and a trend of decreasing rainfall and river discharge have been observed in Chilean Patagonian (41–46° S), including the fjord systems (i.e., Reloncaví, Pitipalena, and Puyuhuapi) [57,58,92]. A recent 12-year time-series analysis showed a decreasing trend in the thickness of the EFW (estuarine freshwater layer, salinity < 11 psu) in Pitipalena Fjord [59]. The EFW layer or its absence and the consequent weakening of the haline stratification in the fjords have been identified as good indicators of favourable conditions for exchange between the fjords and shelf waters. The latter is the habitat of neritic HAB species, such as *Karenia selliformis* and *D. acuta*, adapted to higher salinities and water columns with thermohaline stratification [59,91].

This study highlights the unusual presence of a moderate bloom of oceanic *Karenia* species (>10^5^ cells L^−1^) with the potential to grow further and cause damage to the wild fauna of a protected area, the highly stratified Pitipalena Fjord system. Cell densities >10^5^ of cells L^−1^ of *K. brevis* are considered to be the threshold level above which there may be noxious effects on the marine fauna on the Gulf (of Mexico) coasts of Florida [13]. Development of the bloom described here seemed to be mediated by a combination of wind-driven advection of oceanic waters into the fjord and the favourable conditions offered by the presence of ESW (salinity 21–31) at the surface.

There is a need to understand the link between local climate, hydrodynamics, and species-specific phytoplankton responses to changes in water column structure to develop a risk assessment for the occurrence of oceanic ichthyotoxic species in the Patagonian fjords, where much of the country’s salmon production is located. Salmon farmers in Chilean Patagonia have limited means to detect early warning signals of fish-killing blooms of *Karenia* and adopt mitigation strategies. First, it is difficult to discriminate between harmful and harmless species in multispecific populations. Second, the toxins in Chilean strains of *K. selliformis* have not been chemically identified yet and their occurrence is not targeted in routine monitoring analyses for seafood safety. The monitoring frequency is not sufficient to detect rapid changes caused by wind reversals. Information needs to be complemented with modern operational oceanography tools.

The results here provide helpful information on the environmental conditions and water column structure favouring *Karenia* occurrence. Thermohaline properties in the surface layer in summer can be used to develop a risk index (positive if the EFW layer is thin or absent).

## 4. Materials and Methods

Instantaneous streamflow data from the Palena River corresponding to summer (January to March) 2020 were obtained from the Water Agency website and daily accumulated rainfall data were collected from the Chilean Climate Explorer, Center for Climate and Resilience Research (http://www.cr2.cl/; accessed on 2 March 2023).

### 4.1. Field Sampling of Environmental Conditions and Phytoplankton

On 18–19 March 2020, an intensive 24 h survey was carried out at a fixed station in Pitipalena Fjord (43°47′ S–72°56′ W) (Figure 1C). Vertical profiles of temperature, salinity, dissolved oxygen, and chl *a* fluorescence from the surface to 40 m of depth were obtained every hour with RBR-CTD (conductivity–temperature–depth) profiler (https://rbr-global.com, accessed on 2 March 2023) model Concerto 3 equipped with a Turner Designs CYCLOPS-7 fluorometer (excitation 460 nm, emission 620–715 nm). CTD data processing was carried out with the software provided by the manufacturer and depicted using Ocean Data View software version 5.1 [93].

For the quantitative analysis of phytoplankton, 100 mL seawater samples were collected every 2 h at 2 m intervals from the surface to 20 m and at 25 m and 30 m (n = 169 samples) with a 5-L Niskin bottle and fixed with Lugol’s iodine acidic solution [94].

For toxin analysis, vertical plankton net (20 µm mesh) tows from 20 m deep to the surface were collected every hour. The whole content of the net collector was filtered through Whatman GF/F fibreglass filters (25 mm Ø, 0.7 μm pore size) (Whatman, Maidstone, UK), and filters and filtered material were placed in a cryotube with 1 mL analysis-grade methanol and stored at −20 °C until analysis. Finally, samples for nutrient analysis (NO_3_^−^, NO_2_^−^, PO_4_^3−^, and Si (OH)_4_) were taken every 6 h with 50 mL syringes connected to the spigot of the Niskin bottle at each 4 m interval sample depth (i.e., 0, 4, 8, 12, 16, 20, and 30 m).

### 4.2. Nutrient Analysis

Dissolved inorganic nutrients were analysed with a Seal AA3 AutoAnalyzer according to Grasshoff et al. [95] and seawater analysis was conducted according to the standard method described in Kattner and Becker [96]. Ammonia analyses were omitted due to the impossibility of ensuring that analyses of this labile molecule could be conducted very soon after collection in remote areas in Southern Chile.

### 4.3. Phytoplankton Analysis

Aliquots of 10 mL of each Lugol-fixed bottle sample were left to sediment over 6 h and analysed with an inverted microscope Olympus CX40 (Olympus, Japan), according to Utermöhl [97]. The whole surface of the chamber was scanned at a magnification of 100× (detection limit of 100 cells L^−1^). Given the difficulty of discriminating between different species of the *Karenia* complex in multispecific fixed samples, individuals were identified at the genus level.

### 4.4. Morphometry

Micrographs and cell measurements were made with the inverted microscope Carl Zeiss Primovert (Carl Zeiss, Germany) coupled to a ZEISS Axiocam 208 colour/202 mono camera. The maximum length (L) and width (W) of 50 *Karenia* specimen micrographs were measured with the Image Processing and Analysis in Java (ImageJ) free software [98]. Measurements of cells in apparent binary fission were not included.

### 4.5. Toxin Sample Extraction and Analysis

Tubes with the filtered net tow samples with methanol were centrifuged (20,000× *g*; 20 min) and sonicated with ultrasonic cell disruptor Branson Sonic Power 250 (Danbury, CT, USA). The extract obtained was clarified by centrifugation (20,000× *g*; 15 min) and filtered through 0.20 µm Clarinert nylon syringe filters (13 mm diameter) (Bonna-Agela technologies, Torrance, CA, USA).

LC-HRMS analyses were carried out with a Dionex Ultimate 3000 UHPLC system (Thermo Fisher Scientific, Sunnyvale, CA, USA) coupled to a high-resolution mass spectrometer Q Exactive Focus equipped with an electrospray interphase HESI II (Thermo Fisher Scientific, Sunnyvale, CA, USA). The chromatographic separation was performed following Regueiro et al. [99] with slight modifications, including a shorter reversed-phase HPLC column Gemini NX-C18 (50 mm × 2 mm; 3 μm) with an Ultra Guard column C18 from Phenomenex (Torrance, CA, USA). GYM-A is the only toxin with certified standards commercially available. GYM-A was quantified by comparing peak areas in the chromatograms with those of injected certified reference toxin solutions from the NCR, Canada. The quantification limit of the method was 3 ng mL^−1^.

### 4.6. Satellite Data

The Copernicus Sentinel-2 mission comprises a constellation of two polar-orbiting satellites placed in the same sun-synchronous orbit, phased at 180° to each other. These satellites have a high revisit time (every 2–3 days) at mid-latitudes under cloud-free conditions. High-resolution (10 m) Sentinel-2 images were obtained on 26 March and 31 and 8 April 2020, i.e., 1 week, 12 days, and 2 weeks after the intensive field sampling in PT. Sentinel-2 level-2 data were obtained from the Copernicus open-access website, (https://scihub.copernicus.eu/; accessed on 11 April 2023). The images were projected in the system under reference EPSG: 32718—WGS 84/UTM zone 18S. Natural combined red (B4), green (B3), and blue (B4) colour bands were obtained with a gamma value of 0.3 to optimise image visualisation. The Normalized Difference Chlorophyll Index (*NDCI*) was estimated according to the flowing equation of Mishra and Mishra [100]:(1)$$\;\;\;\;\;\;\;\;\;\;\;\;\;\;\;\;\;NDCI=\frac{\left(Rrs705-Rrs665\right)}{\left(Rrs705+Rrs665\right)}$$
where

*Rrs*705 = remote sensing reflectance at 708 nm (visible and near-infrared NIR).

*Rrs*665 = remote sensing reflectance at 665 nm (red).

Chlorophyll-a (*chl a*) was estimated using Sentinel-2 images and following the OC3 algorithm according to O’Reilly and Werdell [101]:(2)$$\log_{10}chl\;a=\sum_{i=0}^{4}a_{i}X^{i}$$
where

*a_i_* coefficients were obtained from Pahlevan et al. [102].
(3)$$\;X=\log_{10}\frac{max(Rs442,Rs492)}{Rs560}$$

*Rrs*442 = remote sensing reflectance at 442 nm (coastal aerosol).

*Rrs*492 = Idem at 492 nm (blue).

*Rrs*560 = Idem at 492 nm (green).

Sentinel-3 images were used to detect *Karenia* [103] using the difference between red band difference (RBD) and the *Karenia brevis* Bloom Index (*KBBI*) [104]:(4)RBD=nLw(673.75)−nLw(665)

(5)$$KKBI=\frac{nLw\left(673.75)-nLw(665\right)}{nLw\left(673.75)+nLw(665\right)}$$
where

*nL*w (673.75) = normalised water leaving remote at 673.75 nm (Oa09).

*nLw* (665) = Idem at 665 nm (Oa08).

Satellite images from the Group for High-Resolution Sea Surface Temperature (GHRSST) [105] (https://www.ghrsst.org/; accessed on 11 April 2023) were downloaded on the same dates previously indicated. For each daily GHRSST image (1 km resolution), the temperature gradient was calculated as a proxy for the front presence.

## Figures and Tables

**Figure 1 toxins-16-00077-f001:**
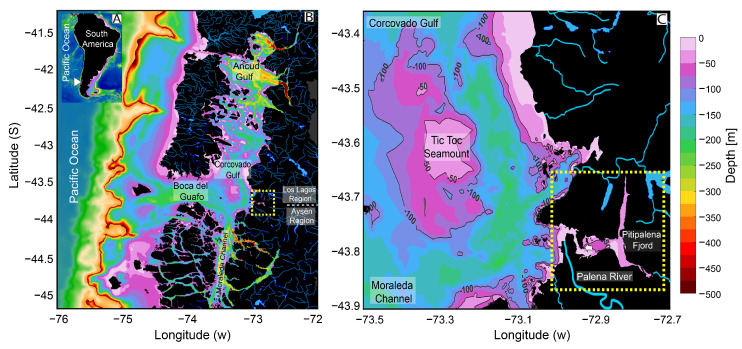
Map of the study area: Chilean coastline and the Patagonian province of Aysén (white arrowhead) (**A**). Northern Patagonia channels and fjords system and their connections to the Pacific Ocean and Pitipalena Fjord (PF) (dotted yellow line frame) (**B**). Location of fixed sampling station (white arrow) for the 24 h study in PF (dotted yellow lines frame) (**C**) and its main freshwater source, the Palena River.

**Figure 2 toxins-16-00077-f002:**
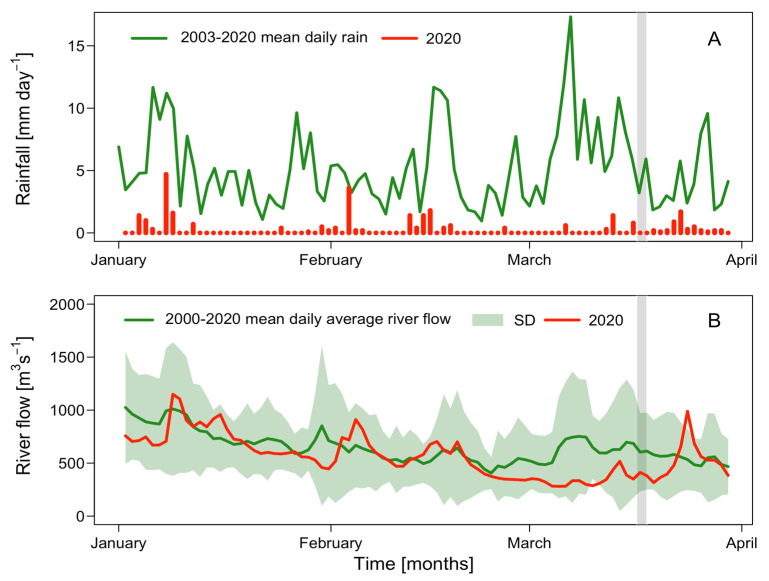
Daily rainfall (mm) (red bars) and historic (2003–2020) daily mean values (green line) in Pitipalena Fjord, January to April 2020 (**A**). Daily flushing rate (m^3^ s**^−^**^1^) (red line), historic daily mean (green line), and standard deviation (shaded area) of the Palena River during the same period (**B**).

**Figure 3 toxins-16-00077-f003:**
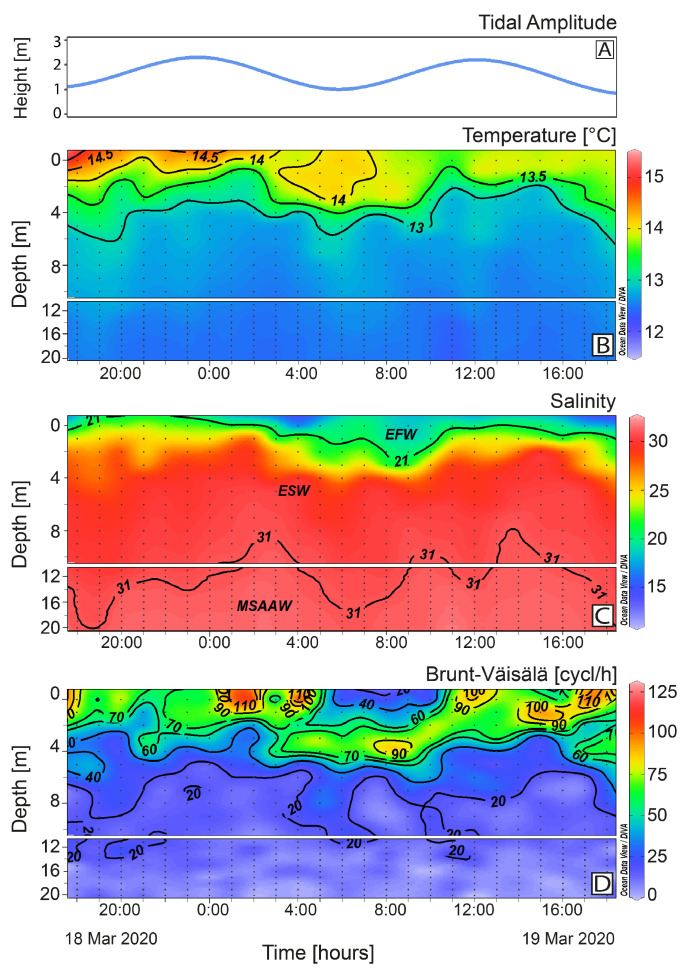
Vertical distribution of tidal amplitude (**A**); hourly vertical distribution (0–20 m) of temperature (°C) (**B**); salinity (**C**); and Brunt–Väisälä frequency (**D**) at the fixed sampling station in Pitipalena Fjord during the 24 h survey in March 2020. Acronyms for water layers stand for: EFW = estuarine fresh water (S = 11–21); ESW = estuarine salty water (S = 21–31); MSAAW = modified subantarctic water (S > 31).

**Figure 4 toxins-16-00077-f004:**
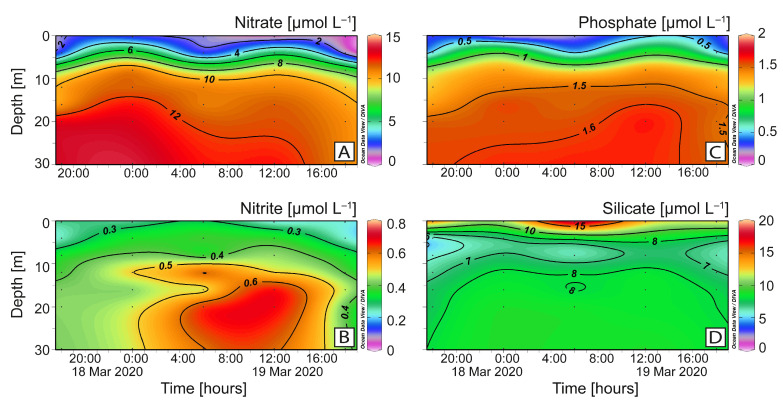
Vertical distribution (0–30 m) every 6 h of dissolved nutrients at the fixed station in Pitipalena Fjord during the 24 h sampling in March 2020. Nitrate (**A**); nitrite (**B**); phosphate (**C**); silicate (**D**).

**Figure 5 toxins-16-00077-f005:**
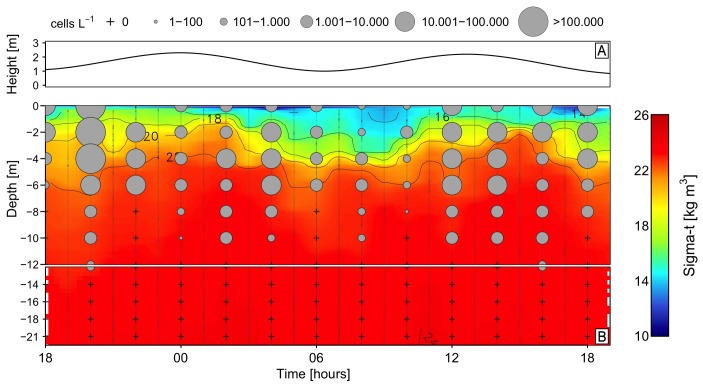
Vertical distribution of tidal amplitude (**A**), *Karenia* sp. cells (**B**), and seawater density (sigma-t) during the 24 h study (18–19 March 2020) at a fixed station in Pitipalena Fjord.

**Figure 6 toxins-16-00077-f006:**
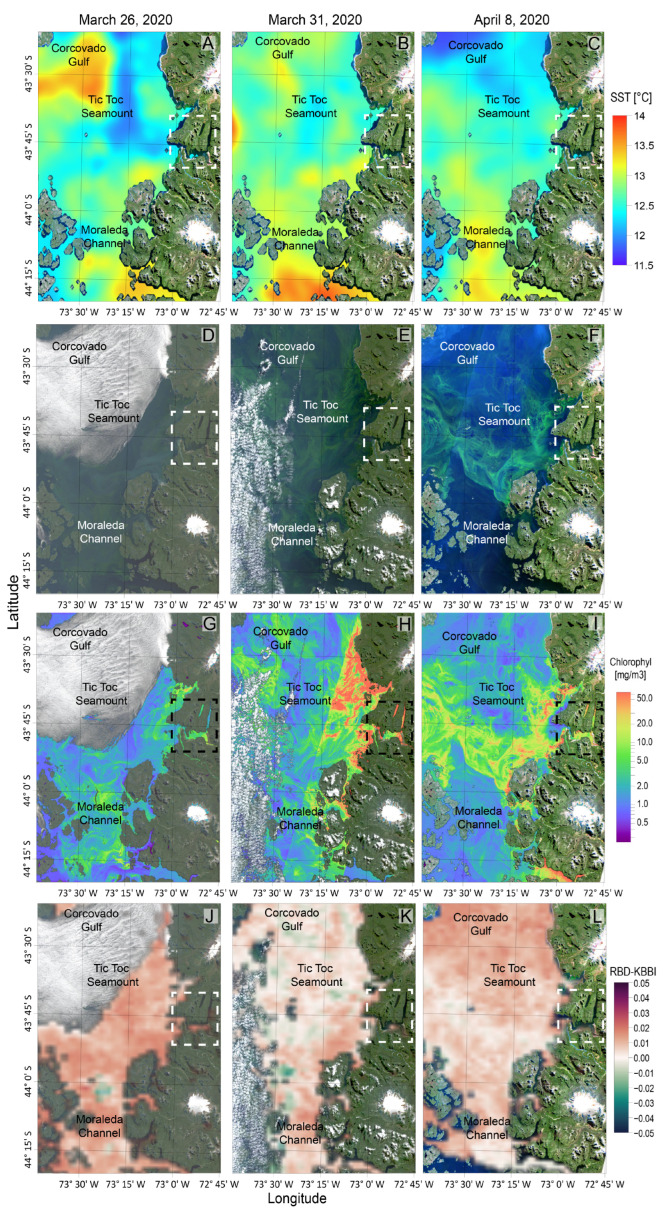
Satellite images of sea surface temperature from GHRSST (1 km) (**A**–**C**); true colour (**D**–**F**) and chl *a* (**G**–**I**); estimates from Sentinel-2 (10 m) and RBD-KKBI (**J**–**L**); and estimates from Sentinel-3 (300 m) observed in the Corcovado Gulf area, NW Patagonia on 26 and 31 March and 8 April 2020.

**Figure 7 toxins-16-00077-f007:**
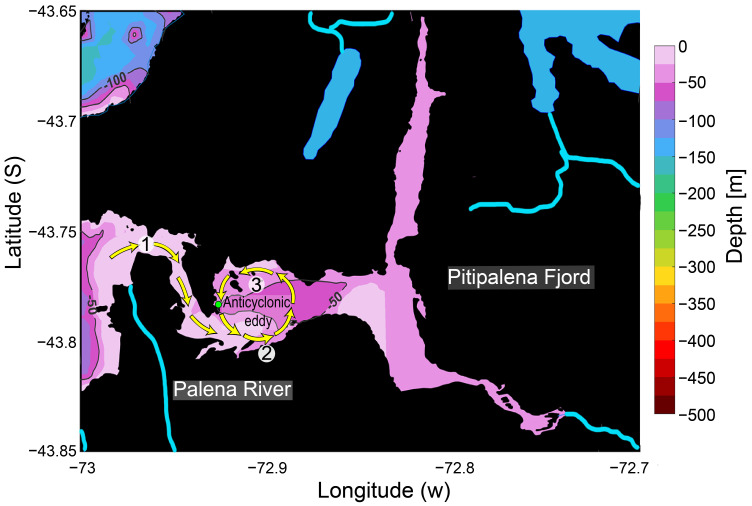
Simplified diagram of the tidal flow pathways (yellow arrows): 1. Entering the channel. 2. Reaching the south coast of the fjord. 3. Forming a particle-retaining an anticlockwise eddy (created from ADCP data in [52]. Green dot indicates the location of the sampling station.

**Table 1 toxins-16-00077-t001:** Monthly values of rainfall and Palena´s river discharge in Pitipalena Fjord in summer 2020 and anomalies in relation to the 17-year (2003–2019) mean.

	Rainfall (mm/month^−1^)			River Discharge (m^3^s^−1^/month^−1^)		
Year	2003–2019 mean	2020	Anomaly	2003–2019 mean	2020	Anomaly
January	155.6 ± 92.4	10.9	−144.7	817.1 ± 242.5	717.6	−99.5
February	124.5 ± 102.1	10.8	−113.7	547.5 ± 170.8	539.9	−7.6
March	172.4 ± 91.61	7.7	−164.7	586.8 ± 221.8	418.6	−168.2

## Data Availability

Data are contained within the article.

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
