# Peer review of "An Unprecedented Bloom of Oceanic Dinoflagellates (Karenia spp.) Inside a Fjord within a Highly Dynamic Multifrontal Ecosystem in Chilean Patagonia"

_toxins, 2024, doi:10.3390/toxins16020077_

Round 1

Reviewer 1 Report

Comments and Suggestions for Authors

Author Response

Greatly appreciate your meticulous report.

REVIEWER 1

Reviewer: This paper describes the patterns of distribution along with environmental, meteorological and hydrographic conditions associated with a bloom of a Karenia species (presumptively K. selliformis or a K. selliformis-like species) in a specific region in Chilean Patagonia that occurred in 2020. This event is put into detailed context with previous occurrences of Karenia and other phytoplankton bloom events, and in some cases fish mortality events, that have occurred in this region (and nearby regions) in the past. Overall, this paper provides a relevant contribution to our general understanding of HAB bloom dynamics, especially in relation to changing ocean conditions and how these changing conditions might lead to bloom events not previously described. Please see specific comments below.

Authors: We really appreciate your detailed review and the relevant comments. We fully agree with practically all of them.

Abstract:

Reviewer: Lines 19-20: This is a minor point, but the use of “chlor a” is not familiar to me as a common way of abbreviating chlorphyll a. I am most familiar with the use of “chl a”. I would suggest making sure to use the most common abbreviation to avoid confusion. It is noticed in other parts of the manuscript, “chl-a” (line 275) and “Chl-a” (line 494) is used. Please be consistent.

Authors: It has been changed to chl a all through the document

Introduction:

Reviewer: Line 47: Again, this is relatively minor, but a point to bring up nonetheless. The regions of the Gulf of Mexico that most frequently experience blooms (the eastern Gulf of Mexico along the coast of Florida, and the western region along the coast of Texas) are sub-tropical, not tropical. The reference provided (ref #5, Brand et al. 2012) does not specifically state a tropical distribution for any of the noted Karenia species. I would suggest either changing “tropical” to “sub-tropical” or, IF there is sufficient evidence (and references to support) that Karenia species have been detected in the tropical region (southern) of the Gulf of Mexico, then add “southern” before “Gulf of Mexico”.

Authors: “Tropical” has been corrected to “subtropical”

Reviewer: Line 51: This states that “Blooms are frequently a mixture of several Kareniaceae…” and again references Brand et al. 2012. That reference does make the statement that “It has been recently discovered that many of these Karenia blooms actually include several species of Karenia”, but no reference or supporting evidence is provided in that reference to support this statement. I would suggest changing “are frequently” to something less emphatic such as “may sometimes” unless additional references that are more specific and supported are provided.

Authors: This bit has been changed citing examples where this multispecific assemblage of Karenia spp. has been observed.

Reviewer: Lines 68-71: References should be provided that support the notion that Karenia spp. engage in all of these carbon-acquiring strategies. The reference provided (#19, Mitra et al. 2016) provides an overview on different strategies in plankton, but does not specifically document these strategies in Karenia spp. Either provide such references or alter the wording to indicate that Karenia may engage in these strategies.

Authors: We added: “and have been found to feed on cyanobacteria (Synechochoccus spp) in laboratory cultures (Jeong et al, 2005).

Reviewer: Lines 100-101 and 101-102: Statements like these should include references.

Authors: Villac et al. 2020 has been cited there. That is an excellent review of the state of the art about Karenia species research and monitoring with a focus on K brevis in the Gulf of Mexico

Reviewer: Line 144: The use of the term “high” in relation to the abundance of Karenia spp. seems arbitrary when applied to 65,000 cells/L, especially when in previous text (line 126) “high” is used to refer to 8,000,000 cells/L. I would suggest replacing “high” with “elevated”

Authors: The term “moderate” has been used for densities below 105 cells/L

Reviewer: Table 1 (page 6). Need to double check some math. The “Anomaly” column of the chart is presumably the 2020 value minus the 2003-2019 mean value. This holds up for all of them except the March River discharge, which has a 2020 value of 418.6 and a 2003-2019 mean value of 586.8, which would result in an anomaly of -168.2, not -171.2 as listed in the chart.

Authors: You are right, there was an error in the calculation which has been corrected and we have also revised the other values.

Results:

Reviewer: Section 2.2 (Line 240): Although I do not consider it essential, it would be helpful if micrographs of the cells were provided. Do the authors have access to any representative micrographs from this work?

Authors: Low resolution micrographs available have been included as Annex 1. The samples are in a bad state now to obtain new images

Reviewer: Line 251: Italicize Karenia.  Authors: Done

Reviewer: Line 258: Why only GYM-A? It seems clear that the species was not known, why was the toxin analysis limited to only one specific toxin? A broader toxicity test (i.e. mouse bioassay or a cytotoxicity assay) would at least have helped determine if there was any level of toxicity.

Authors: GYM-A is the only Karenia-related toxin with commercially available certified standards for chromatographic analyses. The detection of GYM-A by Trefault et al. (2011) in plankton net-hauls collected in offshore waters from Aysén province, suggested that different species of Karenia are part of the autochtonous phytoplankton assemblages in oceanic waters off the Chilean Patagonia. There are no standard methods to analyse GYMs. Concerning the major bloom of Karenia selliformis in Chile in 2018, results from a workshop, and from an international group of experts focused on this issue concluded that:

The massive fauna mortality during K. selliformis bloom events in the Chilean coast cannot be explained by GYMs nor brevetoxins, but can to a large extent be accounted for by the high production of long-chain PUFAs and/or still uncharacterized highly toxic compounds” (Mardones et al., Harmful Algae 2020)”.

Another conclusive summary of the situation with toxin analyses and Karenia blooms appears in the just published IOC Report on fish-killers:

Brevetoxins have so far only been unambiguously detected in Florida strains of Karenia brevis, but their quantitative role in causing fish kills remains to be demonstrated since purified PbTx2,3 exhibited limited ichthyotoxicity against RTgill cells (Dorantes-Aranda et al. 2015) and only in high concentrations. This also applies to brevisulcenals from K. brevisulcata, gymnocin from K. mikimotoi and gymnodimine from K. selliformis, because no suitable analytical methods are available to monitor and assess their concentrations during fish kills in nature.

Reviewer: Line 264: Typo - “Tic Tic” should be “Tic Toc” seamount. Also, in the abstract it is written with a hyphen, and in the other instances it is not. Please ensure the spelling and format are consistent.

Authors: Corrected to “Tic Toc” all through

Reviewer: Figure 6 (page 10): Perhaps there is an explanation for it, but the image in panel H seems to have a fairly clearly demarcated line running through the image (at an angle from ~73o30’ W to ~73o0’ W) that makes it appear that the color intensity interpretation may be a little anomalous, with the color intensity appearing vastly different on the east side of the line vs the west side.

Authors: That’s right. This band corresponds to an area in the central part of the image in which the composition shows a lower intensity, both in the true color (panel E) and in chlorophyll-a (panel H). This area is present at levels 2A and 1C of that period, which are the image sets available to Sentinel users. Due to the high spatial resolution of the sentinel 2 images (10 m), to create a composite image of a large area (e.g. inland sea and Corcovado Gulf) it is necessary to integrate different images. Nevertheless, this "technical artifact" of the satellite image does not affect our interpretation of the spatial patterns in the Corcovado Gulf area and the high chlorophyll at the entrance to the Pitipalena Fjord.

Reviewer: Line 292: There does not seem to be a need to say “other” diatoms. No diatoms had been mentioned yet.

Authors: We agree, “other” has been deleted

Reviewer: Line 298: refers to the satellite image from April 6, but the satellite images provided in Figure 6 include April 8, not April 6.

Authors: Corrected

Discussion:

Reviewer: Lines 309-310: What causes the authors to think that these species have always been there? Certainly, they could have been introduced through a variety of means, including ballast water or even the salmon themselves. I would suggest not making this statement unless a valid argument for asserting they have “always been there” is presented.

Authors: This para. has been re-written. Now there is no statement, but a suggestion (one out of four) to explain why reports didn’t come until the end of the XX Century

Reviewer: Lines 326 and 328: I understand the appropriate occasional use of just the genus name “Karenia”, but in these instances the authors appear to be referring to specific populations, which, according to the manuscript (esp. lines 334-337), is presumptively K. selliformis. I would suggest the addition of sp. or spp. to these specific instances if the exact species could not be determined.

Reviewer: Line 342: I think “Karenia species” would be more appropriate than “Karenia cells”

Authors: The term Karenia has been revised, one by one, and “sp”/”spp”/epithet name added.

Reviewer: Lines 357-358: As with my comment above from Line 258, why was GYM-A the only targeted toxin? The exact Karenia species was not known, so why was the toxin analysis limited to only one specific toxin? The presumption seems to be a K. selliformis-like species, but (as referred to in lines 369-371) previous work has found that K. selliformis from the Chilean Patagonia does not produce GYM-A. So why target that toxin? Lines 360-362 refer to a cytotoxic assay performed in other work. Why was something like that not performed for this study? I understand that there could be constraints, especially since the study was originally targeting Dinophysis and not Karenia, but there should be some justification provided for this decision to limit the toxin analysis. Also, it would be appropriate to include a caveat that this does not necessarily indicate that the bloom was not toxic.

Authors: See explanations above. The cytotoxic assay mentioned on L- 360-362 was applied to cultures of K. selliformis from the region and to other fish-killing species. The results were not very conclusive. A black box of “PUFAs”. This in a multispecific net tow extract would not give a very meaningful results and could correspond to anything, even to accompanying organic matter.

Reviewer: Line 362: Again, the genus name “Karenia” is used alone without sp. or spp. As mentioned earlier in this review, there can be appropriate uses for the genus only, but this paper seems to overuse the genus-only. In this instance, the authors are referring to past efforts at quantifying icthyotoxicity, which presumably would have been conducted on a single strain, isolate, or natural population. And certainly, there could be large differences between species within the genus. As written, it comes across as the entire genus shows higher icthyotoxicity than other HAB species. The use of sp. or spp. would be appropriate.

Authors: We added “This K. selliformis strain..”

Reviewer: Line 395-397: This statement is unclear. Please revise or eliminate this line.

Authors: “The coupling with the tides suggests the population was advected from oceanic waters off PF through Boca del Guafo”  This inaccurate statement has been deleted

Reviewer: Lines 400-401 and 402-405: There seems to be an over-reliance on the Brand et al. 2012 paper for referencing certain aspects of Karenia distribution, biology and behavior that were originally reported in other works. While it is certainly acceptable to use a review-type paper for some basic information, it is best to go to original sources for specific information, such as evidence for DVM and process-linked depth aggregation.

Authors: We apologize for excessive use of a review paper. We have provided now more references corresponding to the original sources.

Reviewer: Lines 426-427: “low-salinity” vs “high salinity” Please be consistent in the use or hyphens.

Authors: Corrected. All without hyphens

Reviewer: Lines 470-472: This sentence is unclear. Was there something left out at the beginning of the sentence?

Authors: This is now rephrased in a new sentence: “Additional fronts are formed by topographically induced upwelling around the Tic Toc Seamount (Aguirre et al., 2012; Pérez-Santos et al., 2019)

Reviewer: Line 517: misspelling of “outline”; Line 535: a parenthesis is missing

Authors: both errors have been corrected

Reviewer: Lines 550-551: These should be one paragraph. No need to split.

Authors: They have been put in the same paragraph

  1. Reviewer: Lines 536-540: I think I understand what this paragraph is trying to covey, but it is a little “muddied” in the way it is written. Could this rewritten to clarify the point?

  1. Reviewer: Lines 554-555: This sentence is unclear. Is something missing after “…wind driven…”?

  1. Reviewer: Lines 556-560: This closing paragraph could be better considered in the context of the entire paper. First, it is stated that the thermohaline properties could be used as a “risk index” but such an index was not previously discussed or described in the manuscript. Maybe better to suggest that this type of information could be used to develop a risk index. Also, the last sentence recommends a combination of satellite imagery and ground truthing, but this seems an unnecessary statement as satellite imagery for detecting HABs always needs ground truthing and there is not really any discussion as to why, specifically, the authors suggest this here. Also, there are several typos in the last line (line 560).

Authors: (1-3) The whole discussion has been reviewed and reorganized taking into account your relevant advice. The last section lines now read:

“A recent 12-year time-series analysis showed a decreasing trend in the thickness of the EFW (estuarine freshwater layer, salinity <11 psu) in Pitipalena fjord [60]. EFW layer or its absence, and the consequent weakening of the haline stratification in the fjords have been identified as good indicators of favourable conditions for exchange between the fjords and shelf waters. The latter are the habitat of neritic HAB species, such as Karenia selliformis and D. acuta, adapted to higher salinities and water columns with thermohaline stratification [60,92].

This study highlights the unusual presence of a moderate bloom of oceanic Karenia species in a highly stratified system as PF, possibly mediated by a combination of wind-driven advection of oceanic waters into the fjord, and the favorable conditions offered by the presence of ESW (salinity 21- 31) at the surface.

There is a need to understand the link between local climate, hydrodynamics and species-specific phytoplankton responses to changes in water column structure to develop risk assessment of the occurrence of oceanic ichthyotoxic species in the Patagonian fjords where much of the country's salmon production is located. Salmon farmers in Chilean Patagonia have limited means to detect early warning signals of fish killing blooms of Karenia and adopt mitigation strategies. First, it is difficult to discriminate harmful and harmless species in multispecific populations. Second, the toxins in Chilean strains of K. selliformis have not been chemically identified yet and their occurrence is not targeted in the routine monitoring analyses for seafood safety. Monitoring frequency is not sufficient to detect rapid changes caused by wind reversals. Information needs to be complemented with modern operational oceanography tools. Results here provide helpful information on the environmental conditions and water column structure favouring Karenia occurrence. Thermohaline properties in the surface layer in summer can be used to develop a risk index (positive if the EFW layer is thin or absent). “

Methods:

Reviewer: Line 654: I believe “imagenes” is a typo. Authors: Done

Appendix A:

Reviewer: Figure A2: In the text, this figure is referred to after the statement that “Targeted toxin analyses by LC-HRMS showed no traces of GYM-A either in plankton tows or in filtered bottle 258 sample extracts” (Lines 257-259). However, it is not clear what Figure A2 is actually showing us. The figure caption indicates it is the retention time and mass spectra of GYM-A, but the text (from lines 257-259) would indicate that it is showing us results from a sample that did NOT contain GYM-A. Please clarify what this figure is presenting.

Authors: Our mistake. Two images should have been there: one corresponding to the injected sample extract and the other to the certified GYM-A standard. Now is corrected.

Reviewer 2 Report

Comments and Suggestions for Authors

The manuscript describes a bloom of Karenia spp in summer 2020. It was detected in samples from a 24-h study focused on Dinophysis spp. in the confines of the Pitipalena-Añihue Marine Protected Area. This ephemeral event occurred at the beginning of the COVID-19 lockdown, when monitoring activities were relaxed. Some salmon mortalities were related to the deaths of other fish species such as Margalefidinium polykrikoides. This event was also characterized by negative anomalies in rainfall and river flow and a severe drought in the month of March. The cells of Karenia spp. were in a warmer (14-15°C) surface layer of estuarine salt water that was diluted with the ebbing tide. The authors propose a conceptual circulation model to explain the hypothetical retention of Kareneia bloom by a coastal-generated eddy coupled to semidiurnal tides at the mouth of the Pitipalena Fjord. The showed satellite images confirmed the decline of Karenia spp. but they gave evidence of multifrontal dynamic patterns of temperature and Chlorophyll-a distribution. Following this theory, decreasing trends in precipitation and river discharge are expected to favor the advection of oceanic species that can cause damage to fish.

On the one hand, this is the First report of a Karenia selliformis-like bloom (with no Gymnodimines) inside a North Patagonia fjord and description of its circadian variability at Corcovado Gulf, adjacent to the fjords, where flagellates accumulate, which are potential sources of fish-killing HABs.

On the other hand, the bloom was ephemeral. Furthermore, little information is included on the correct identification of the microalgae: photographs, molecular studies, sequencing, etc. This information should also be included in the article.

Regarding toxin content, only GYM-A was monitored among the many toxins produced by Karenia. Nor is there a total toxicity analysis of the extract in cells or in mice that indicates the true toxicity of the strain.

In Materials and Methods, it is indicated that the samples were taken every 2 hours but the total number of samples or the total sampling period is not indicated. It should be known to see if the number of samples is significant.

Figure A1 corresponds to a certified GYM-A reference? If so, it contributes very little to the discussion and should be eliminated.

Figure 3A. The significant graph for Karenia spp. 10-23 Feb has very poor resolution and should be improved.

Author Response

REVIEWER 2

Authors: Thanks for your time!. See our answers below

The manuscript describes a bloom of Karenia spp in summer 2020. It was detected in samples from a 24-h study focused on Dinophysis spp. in the confines of the Pitipalena-Añihue Marine Protected Area. This ephemeral event occurred at the beginning of the COVID-19 lockdown, when monitoring activities were relaxed. Some salmon mortalities were related to the deaths of other fish species such as Margalefidinium polykrikoides. This event was also characterized by negative anomalies in rainfall and river flow and a severe drought in the month of March. The cells of Karenia spp. were in a warmer (14-15°C) surface layer of estuarine salt water that was diluted with the ebbing tide. The authors propose a conceptual circulation model to explain the hypothetical retention of Kareneia bloom by a coastal-generated eddy coupled to semidiurnal tides at the mouth of the Pitipalena Fjord. The showed satellite images confirmed the decline of Karenia spp. but they gave evidence of multifrontal dynamic patterns of temperature and Chlorophyll-a distribution. Following this theory, decreasing trends in precipitation and river discharge are expected to favor the advection of oceanic species that can cause damage to fish.

On the one hand, this is the First report of a Karenia selliformis-like bloom (with no Gymnodimines) inside a North Patagonia fjord and description of its circadian variability at Corcovado Gulf, adjacent to the fjords, where flagellates accumulate, which are potential sources of fish-killing HABs.

Reviewer: On the other hand, the bloom was ephemeral. Furthermore, little information is included on the correct identification of the microalgae: photographs, molecular studies, sequencing, etc. This information should also be included in the article.

Authors: The word “ephemeral” was not appropriate, because we do not know what happened in this part of theh fjord before and after out observations

Reviewer: Regarding toxin content, only GYM-A was monitored among the many toxins produced by Karenia. Nor is there a total toxicity analysis of the extract in cells or in mice that indicates the true toxicity of the strain.

Authors: GYM-A is the only Karenia-related toxin with commercially available certified standards for chromatographic analyses. This toxin was detected in plankton net tow extracts during a cruise in Chilean shelf waters (Trefault et al., 2011). There are no standard methods to analyse GYMs Concerning the major blooms of Karenia selliformis in Chile in 2018 and 2020, results from a workshop, and from an international group of experts focused on this issue concluded that:

The massive fauna mortality during K. selliformis bloom events in the Chilean coast cannot be explained by GYMs nor brevetoxins but can to a large extent be accounted for by the high production of long-chain PUFAs and/or still uncharacterized highly toxic compounds” (Mardones et al., Harmful Algae 2020)”.

Another conclusive summary of the situation with toxin analyses and Karenia blooms appears in the just published IOC Report on fish-killers:

Brevetoxins have so far have only been unambiguously detected in Florida strains of Karenia brevis, but their quantitative role in causing fish kills remains to be demonstrated since purified PbTx2,3 exhibited limited ichthyotoxicity against RTgill cells (Dorantes-Aranda et al. 2015) and only in high concentrations. This also applies to brevisulcenals from K. brevisulcata, gymnocin from K. mikimotoi and gymnodimine from K. selliformis, because no suitable analytical methods are available to monitor and assess their concentrations during fish kills in nature.

In summary, we know very little about the toxins/allelochemicals/ROS/PUFA’s in K. selliformis and other Kareniaceans in Chile and in the rest of the world. It is not surprising the poor information on toxins obtained after revisiting data from a cruise where Karenia species were accidentally detected in lugol-fixed samples weeks after being collected. Still, given the exceptionality of a bloom (>105 cell/L) inside a fjord and the other unique observations on accompanying oceanographic conditions, we thought it was worth taking lessons from it and publish our interpretation and recommendations for an early warning of these events.

Reviewer: In Materials and Methods, it is indicated that the samples were taken every 2 hours but the total number of samples or the total sampling period is not indicated. It should be known to see if the number of samples is significant.

Authors: You are right. We have added the detail of the sampling time and the total number of bottle samples and net trawls.

Reviewer: Figure A1 corresponds to a certified GYM-A reference? If so, it contributes very little to the discussion and should be eliminated.

Authors: We included now the missing graph corresponding to our sample analysis next to the standard graph. We wish to provide evidence of the analyses performed.

Reviewer: Figure A3. The significant graph for Karenia spp. 10-23 Feb has very poor resolution and should be improved.

Authors: The graph for Karenia spp. distribution on 10-23 Feb has been improved.

Reviewer 3 Report

Comments and Suggestions for Authors

This is excellent work. A minor suggestion is to pare back the priority claims, since they are not necessary for the zcience to be interesting and important, and it would require a systematic review to demonstrate that no previous studies on the topic exist. An easy fix is to say "As far as we are aware, this is the first..." or just omit the priority claims as unnecessary.

Author Response

Authors: Thanks you for your positive comments about our work. We followed your recommendation.

Reviewer 4 Report

Comments and Suggestions for Authors

This MS reported Karenia selliformis-like species variation in the outer reaches of the Piti palena-Añihue Marine Protected Area. Other environmental factors were also monitored. The MS provided a plenty of data, but the science, the explanation of the observed results are not good. I would recommend it not be accepted in its current form.

1 Kareia spp are difficult to be identified by morphological characters, but easy to using molecular method. As there are aquatic animal mortality, the accurate identification of phytoplankton species is needed.

2 A cell density of Karenia sp. at 140000 cells/L does not reach to the bloom density.

3. The part of Abstract is complicated, what's the purpose, and the main finding of this work?

4. The organization of "Introduction" is not good. The authors listed a lot of work, but lack of digesting.

5. The part of "Discussion", please focus on your results, not just list others' work.

Author Response

REVIEWER 4

Reviewer: This MS reported Karenia selliformis-like species variation in the outer reaches of the Pitipalena-Añihue Marine Protected Area. Other environmental factors were also monitored. The MS provided a plenty of data, but the science, the explanation of the observed results are not good. I would recommend it not be accepted in its current form.

Authors: The manuscript went through a major revision.

Reviewer: Karenia spp are difficult to be identified by morphological characters, but easy to using molecular method. As there are aquatic animal mortality, the accurate identification of phytoplankton species is needed.

Authors: There are no molecular probes designed yet for any Karenia species from Chile. We hope to have them soon

Reviewer: A cell density of Karenia sp. at 140000 cells/L does not reach to the bloom density.

Authors:  Densities equal or higher than 105 cells/L are the threshold to start causing damage to the wild fauna. We have written this already and added the relevant references from Florida studies with K. brevis

Reviewer: The part of Abstract is complicated, what's the purpose, and the main finding of this work?

Authors: The abstract has been modified and objectives and achievements expressed more clearly.

Reviewer: The organization of "Introduction" is not good. The authors listed a lot of work, but lack of digesting.

Authors: The whole introduction (L. 47 to 204) has been reorganized in order to: i) be more concise and focused; ii) explain more clearly the work objectives and iii) the reasons of bad morphological and toxin profiles descriptions, partly due to the fact that material and methods prepared for the cruise were not planned to deal with what we unexpectedly found.

Reviewer: The part of "Discussion", please focus on your results, not just list others' work.

Authors Discussion has also been revised and made more concise. But comparison with other system and parts of the world have been kept.

Reviewer 5 Report

Comments and Suggestions for Authors

This manuscript reports a highly descriptive study regarding the occurrence of a bloom of potentially toxic dinoflagellates Karenia spp. The study is carried out in Chilean Patagonia, which is a highly productive oceanic region. Although this study presents important discussions and describes the environmental factors that may be responsible for bloom development, new data regarding the Karenia blooms is limited. The main results of this study are the vertical distribution of Karenia. In the opinion of this reviewer, the manuscript to become a research article should be extensively revised, synthesized, and the objectives of the study must be clearly identified (line 182 -188 should be revised).  

Some minor issues:

·       - Line 17: Karenia

·       - Figure 3A. Indicate on the top of the plate A “Tidal Amplitude”

·       - Although the authors indicate that Karenia species may produce “brevetoxins, gymnodimines, gymnocines, polyunsaturated fatty acids (PUFAs), sterols and other toxins of unknown mechanisms of action” only GYM-A was determined in this study. Please justify.

·       - Karenia cells were not identified at species level. Please justify.

Author Response

REVIEWER 5

Many thanks for the time invested in our paper

Reviewer: This manuscript reports a highly descriptive study regarding the occurrence of a bloom of potentially toxic dinoflagellates Karenia spp. The study is carried out in Chilean Patagonia, which is a highly productive oceanic region. Although this study presents important discussions and describes the environmental factors that may be responsible for bloom development, new data regarding the Karenia blooms is limited. The main results of this study are the vertical distribution of Karenia. In the opinion of this reviewer, the manuscript to become a research article should be extensively revised, synthesized, and the objectives of the study must be clearly identified (line 182 -188 should be revised). 

Authors: The whole introduction (L. 47 to 204) has been reorganized in order to: i) be more concise and focused; ii) explain more clearly the work objectives and iii) the reasons of bad morphological and toxin profiles descriptions, partly due to the fact that material and methods prepared for the cruise were not planned to deal with what we unexpectedly found.

Reviewer: Line 17: Karenia

Authors: Corrected

Reviewer: Figure 3A. Indicate on the top of the plate A “Tidal Amplitude”

Authors: Tidal Amplitude” has been added on the top of the panel A

Reviewer: Although the authors indicate that Karenia species may produce “brevetoxins, gymnodimines, gymnocines, polyunsaturated fatty acids (PUFAs), sterols and other toxins of unknown mechanisms of action” only GYM-A was determined in this study. Please justify.

Authors: GYM-A is the only Karenia-related toxin with commercially available certified standards for chromatographic analyses. The detection of GYM-A by Trefault et al. (2011) in plankton net-hauls collected in offshore waters from Aysén province, suggested that different species of Karenia are part of the autochtonous phytoplankton assemblages in oceanic waters off the Chilean Patagonia.. There are no standard methods to analyse GYMs

Concerning the major blooms of Karenia selliformis in Chile in 2018 and 2020, results from a workshop, and from an international group of experts focused on this issue concluded that:

The massive fauna mortality during K. selliformis bloom events in the Chilean coast cannot be explained by GYMs nor brevetoxins, but can to a large extent be accounted for by the high production of long-chain PUFAs and/or still uncharacterized highly toxic compounds” (Mardones et al., Harmful Algae 2020)”.

Another conclusive summary of the situation with toxin analyses and Karenia blooms appears in the just published IOC Report on fish-killers:

Brevetoxins have so far have only been unambiguously detected in Florida strains of Karenia brevis, but their quantitative role in causing fish kills remains to be demonstrated since purified PbTx2,3 exhibited limited ichthyotoxicity against RTgill cells (Dorantes-Aranda et al. 2015) and only in high concentrations. This also applies to brevisulcenals from K. brevisulcata, gymnocin from K. mikimotoi and gymnodimine from K. selliformis, because no suitable analytical methods are available to monitor and assess their concentrations during fish kills in nature.

Reviewer: Karenia cells were not identified at species level. Please justify.

Authors: Species of Karenia are naked (or unarmoured) dinoflagellates, i.e., they lack the rigid cover of cellulose plates characteristic in thecate dinoflagellates. For this reason, their real shape needs to be observed in vivo under the light microscope, and samples to be fixed with glutaraldehyde for scanning electron microscopy. Some morphological features can be roughly described from lugol fixed samples, but most times these are not enough for accurate identification at species level unless it was a frequently observed strain and the local experts were familiar with it. Molecular tools are the solution in laboratories which have them designed for their local strains